# Clofoctol inhibits SARS-CoV-2 replication and reduces lung pathology in mice

**Sandrine Belouzard[1‡], Arnaud Machelart[1‡], Valentin Sencio[1☉], Thibaut Vausselin[1,2☉], Eik Hoffmann[1], Nathalie Deboosere[1,3], Yves Rouillé[1], Lowiese Desmarets[1], Karin Séron[1], Adeline Danneels[1], Cyril Robil[1], Loic Belloy[2], Camille Moreau[2], Catherine Piveteau[4], Alexandre Biela[4], Alexandre Vandeputte[1,3], Séverine Heumel[1], Lucie Deruyter[1], Julie Dumont[3,4], Florence Leroux[3,4], Ilka Engelmann[5], Enagnon Kazali Alidjinou[5], Didier Hober[5], Priscille Brodin[1,3‡], Terence Beghyn[2‡], François Trottein[1‡], Benoit Deprez[3,4‡]\*, Jean Dubuisson[1‡]\***

**1** Univ Lille, CNRS, INSERM, CHU Lille, Institut Pasteur de Lille, Center for Infection and Immunity of Lille, Lille, France, **2** APTEEUS, Campus Pasteur Lille, Lille, France, **3** Univ. Lille, CNRS, Inserm, CHU Lille, Institut Pasteur de Lille, Plateformes lilloises en biologie et santé, Lille, France, **4** Univ Lille, Inserm, Institut Pasteur de Lille, Drugs and Molecules for Living Systems, Lille, France, **5** Univ Lille, CHU Lille, Laboratoire de Virologie, Lille, France

☉ These authors contributed equally to this work.
‡ SB and AM are equally contributing first authors. PB, TB, FT, BD and JD are equally contributing senior authors.
\* benoit.deprez@univ-lille.fr (BD); jean.dubuisson@ibl.cnrs.fr (JD)

**Data Availability Statement:** All relevant data are within the manuscript and its Supporting Information files.

## Abstract

Drug repurposing has the advantage of shortening regulatory preclinical development steps. Here, we screened a library of drug compounds, already registered in one or several geographical areas, to identify those exhibiting antiviral activity against SARS-CoV-2 with relevant potency. Of the 1,942 compounds tested, 21 exhibited a substantial antiviral activity in Vero-81 cells. Among them, clofoctol, an antibacterial drug used for the treatment of bacterial respiratory tract infections, was further investigated due to its favorable safety profile and pharmacokinetic properties. Notably, the peak concentration of clofoctol that can be achieved in human lungs is more than 20 times higher than its $IC_{50}$ measured against SARS-CoV-2 in human pulmonary cells. This compound inhibits SARS-CoV-2 at a post-entry step. Lastly, therapeutic treatment of human ACE2 receptor transgenic mice decreased viral load, reduced inflammatory gene expression and lowered pulmonary pathology. Altogether, these data strongly support clofoctol as a therapeutic candidate for the treatment of COVID-19 patients.

## Author summary

Antivirals targeting SARS-CoV-2 are sorely needed. In this study, we screened a library of approximately 2000 drug compounds that have been used or are still used in the clinics. Among them, we identified clofoctol as an antiviral against SARS-CoV-2. This molecule is an antibacterial drug used for the treatment of bacterial respiratory tract infections and it was further investigated due to its safety profile and its properties to accumulate in the lungs. We further demonstrated that, in vivo, this compound reduces inflammatory gene

**Funding:** This work was supported by the Institut Pasteur de Lille (to JeD and BD), the Fondation pour la Recherche Médicale (FRM to JeD) and the Agence Nationale de la Recherche (ANR) (Project FRM_ANR Flash 20 ANTICOV to JeD), the Centre National de la Recherche Scientifique (CNRS: COVID and ViroCrib programs to JeD) and the I-SITE ULNE Foundation (I-Site_Covid20_ANTI-SARS2 to JeD) and the Conseil Régional Hauts-de-France (THERAPIDE grant N°20005467 to BD). We also received sponsor support from LVMH (to BD), fondation Rotary (to BD), Vinted (to BD), Crédit Mutuel Nord Europe (to BD), Entreprises et Cités (to BD), AG2R (to BD), DSD Système (to BD), M comme Mutuelle (to BD), Protecthoms (to BD), RBL Plastiques (to BD), Saverglass (to BD), Brasserie 3 Monts (to BD), Coron Art (to BD). EH received support from the I-SITE ULNE Foundation (ERC Generator Grant). The platform used in this work was supported by the European Union (ERC-STG INTRACELLTB grant 260901), the ANR (ANR-10-EQPX-04-01), the "Fonds Européen de Développement Régional" (Feder) (12001407 [D-AL] EquipEx ImagInEx BioMed), CPER-CTRL (Centre Transdisciplinaire de Recherche sur la Longévité) and the Région Hauts-de-France (convention 12000080). The funders had no role in study design, data collection and analysis, decision to publish, or preparation of the manuscript.

**Competing interests:** I have read the journal's policy and the authors of this manuscript have the following competing interests:European Patent Application Serial No. EP20305633.8, entitled "Compound and method for the treatment of coronaviruses" related to this work was filed on 10 June 2020. Authors TB, LB, CM, SB, PB, ND, BD, JeD, EH, AM, YR and TV of this manuscript are inventors of the patent.

expression and lowers pulmonary pathology. The antiviral and anti-inflammatory properties of clofoctol, associated with its safety profile and unique pharmacokinetic properties make a strong case for proposing clofoctol as an affordable therapeutic candidate for the treatment of COVID-19 patients.

## Introduction

The coronavirus disease 2019 (COVID-19) is having a catastrophic impact on human health as well as on the global economy, and it will continue to affect our lives for years to come [1]. This extraordinary situation led to the rapid development of safe and effective vaccines that have now been deployed at unprecedented scale. While COVID-19 vaccines have demonstrated their essential role in reducing hospitalizations, they do not control virus transmission, and we still lack affordable efficient therapies against SARS-CoV-2. Antivirals are indeed urgently needed to treat COVID-19 patients who have not yet been vaccinated and as a therapeutic approach to treat vaccinated people poorly protected due to waning immunity. An affordable antiviral treatment that could be administered at a large scale to all diagnosed patients at early stage could also contribute to lowering viral load in the respiratory tract and disease transmission. Repurposing clinically evaluated drugs can potentially offer a fast track for the rapid deployment of treatments for this kind of emerging infectious disease. However, the first attempts of targeted repurposing strategies to treat COVID-19 patients have led to disappointing results [2]. As an alternative approach, large-scale screening of clinically approved drugs through a carefully designed evaluation cascade may identify additional unanticipated therapeutic options that can be positioned for accelerated clinical evaluation [3–5]. Here, we developed a high-content screen (HCS) using the Apteeus drug library (TEELibrary), a comprehensive collection of 1,942 approved drugs, to identify molecules that exhibit antiviral activity against SARS-CoV-2. Clofoctol was selected based on its antiviral potency associated with favorable pharmacokinetic properties in human. Its further validation in a small-animal model makes it a promising candidate treatment for clinical evaluation in COVID-19 patients.

## Results

### HCS screening of a library of approved drugs

The screen was performed in Vero-81 cells, an African green monkey kidney cell line highly permissive to SARS-CoV-2 infection [6]. The read out was based on the cytopathic effect (CPE) of the virus as measured by propidium iodide (PI) incorporation into the nuclei of dying cells and cell quantification by nuclei staining with Hoechst 33342 (S1A Fig and S1 Table). Assay parameters, including cell seeding density, multiplicity of infection (MOI) and time points, were optimized in Vero-81 cells by measuring virus-induced CPE in a 384-well format.

To assess reproducibility of the optimized assay in a HCS configuration, we initially evaluated the effect of chloroquine (CQ), previously reported to have antiviral activity against SARS-CoV-2 in Vero cells [7]. This compound is an effective inhibitor of coronavirus entry into host cells through the endocytic pathway, however because of its lack of effect on the TMPRSS2-mediated pathway, chloroquine has been shown to be ineffective to treat COVID-19 patients. Nonetheless, this enabled us to benchmark the dynamic range of the assay with a reliable positive control. Robustness was then assessed by calculating the strictly standardized mean difference (SSMD) of each plate, with a mean of 6.87 (±2.19) for all plates (S1B Fig). We

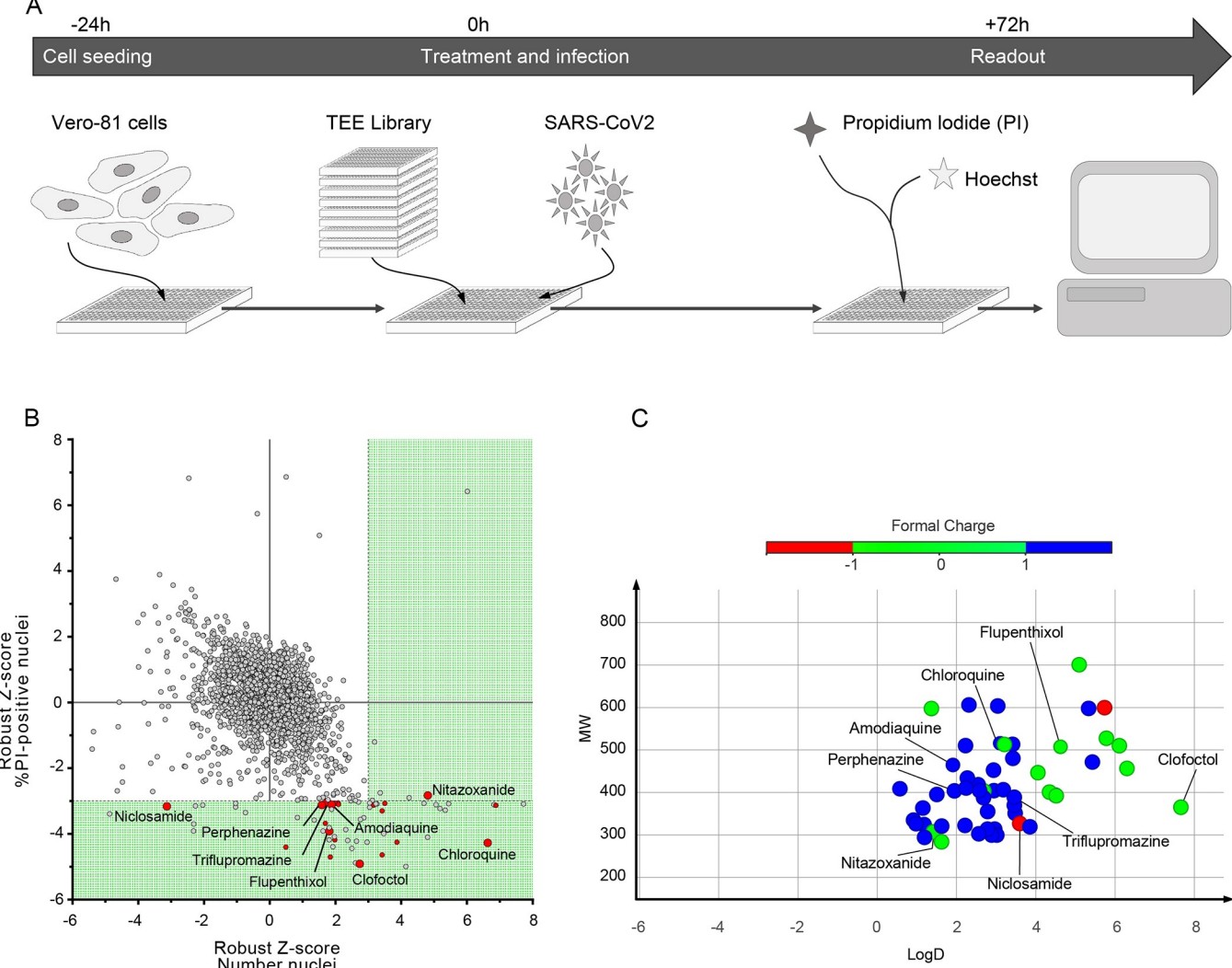

**Fig 1. HCS screen of Apteeus TEElibrary for the identification of anti-SARS-CoV-2 compounds. A,** Workflow overview of the screening process. **B,** Dot-plot representations of all compounds tested based on their robust-Z-score for both the numbers of nuclei and the percentages of PI-positive nuclei. Dotted lines are indicative of the thresholds chosen for hit selectivity (within the green area). **C,** Dot-plot representation according to the molecular weight and the LogD of the compounds of interest. Dots are color-coded based on the ionization state at physiological pH.

then used this experimental design to screen our drug library (Fig 1A) using a non-cytotoxicity concentration of 15 μM for most compounds in the presence of low viral input (MOI = 0.01), in order to capture multicycle replication with an extended end-point measurement at 72h post-infection [4]. Indeed, at this time-point, a major cytopathic effect could be observed with a strong decrease in cell number at the low MOI used in our assay. For each compound, a robust Z-score normalized to the median of each plate was calculated for both SARS-CoV-2-induced CPE related readouts (PI measurement and Hoechst 33342 staining).

Compounds exhibiting the highest levels of CPE inhibition were initially selected. Of the 1,942 tested compounds, 57 were identified to significantly decrease PI incorporation (robust Z-core < -3) or to increase the number of cells as measured with Hoechst 33342 staining (robust Z-score > 3) (S2 Table). Among these compounds, CQ, nitazoxanide, amodiaquine, triflupromazine and niclosamide were previously identified in other studies [3,7–9] (Fig 1B). It

is worth noting that the majority of the selected compounds are basic molecules like CQ, amodiaquine, fluphenazine, trifluoperazine and triflupromazine (Fig 1C). Such compounds are known to affect the endosomal internalization pathway by accumulating inside the endosomes, modifying their pH and therefore inhibiting SARS-CoV-2 by blocking its endosomal entry.

We assessed the activity of the 57 identified hits in dose-response curve (DRC) analyses using the same setting as in the screen (S2 Table). Of these, 21 showed dose-related inhibition of the PI incorporation upon infection (N = 2).

Coronaviruses, including SARS-CoV-2, can use two different routes to enter their target cells. They either enter cells by endocytosis and release their genome into the cytosol after fusion of their envelope with an endosomal membrane, or they fuse their envelope directly with the plasma membrane. This latter entry route is triggered by the cell surface protease TMPRSS2 which is not expressed in Vero-81 cells [10]. Previously identified anti-SARS-CoV-2 compounds, like CQ and hydroxy-CQ, were shown to only block the endocytic entry route [10]. An additional validation on Vero-81-TMPRSS2 cells was therefore performed to discard compounds that only block the endocytic route of SARS-CoV-2 entry.

Of the 21 molecules validated twice in Vero-81 cells, the most interesting ones were retained based on a preliminary evaluation of their risk/benefit ratio in the clinic. This evaluation included a comparison of the *in vitro* potency to plasma exposure at the approved dose. Therefore, only 8 out of the 21 compounds were tested in Vero-81-TMPRSS2 cells (S2 Table). Only 3 of them exhibited a dose-dependent antiviral activity against SARS-CoV-2 in the presence or absence of TMPRSS2, indicating an antiviral effect irrespective of the entry route. These three compounds are perphenazine, nitazoxanide, and a third one, called clofoctol that was not reported by others. More importantly, clofoctol is well distributed in tissues, particularly in lungs where its concentration is twice higher than in plasma [11]. Altogether, these observations prompted us to further characterize the anti-SARS-CoV-2 properties of clofoctol.

### *In vitro* validation of the antiviral activity of clofoctol

To further confirm the antiviral activity of clofoctol, SARS-CoV-2 genomic replication was measured by quantitative RT-PCR. In this assay, clofoctol exhibited an equal potency in Vero-81 cells and in Vero-81-TMPRSS2 with $IC_{50}$ of 12.41 μM and 13.51 μM, respectively (Fig 2A). To validate the specificity of its antiviral activity, its potential cytotoxic effect in cell culture was determined in an MTS viability assay. As shown in Fig 2B, after 24h of treatment, no cytotoxic effect was observed at concentrations below 40 μM, indicating that the decreased SARS-CoV-2 replication in the presence of clofoctol was not due to a cytotoxic effect of the compound. To further characterize the antiviral activity of clofoctol, its effect on the production of infectious progeny virions was also quantified. As shown in Fig 2C, a dose-dependent decrease of infectious virus production was observed in these experimental conditions, confirming the antiviral effect of clofoctol against SARS-CoV-2 with an $IC_{50}$ of 9.3 μM and 11.59 μM in Vero-81 and Vero-81-TMPRSS2 cells, respectively.

Vero cells are the cells of choice to efficiently grow SARS-CoV-2 in culture and therefore to screen large libraries of compounds for rapid identification of antivirals. However, as these cells are from monkey origin, the human cell line Calu-3, derived from a lung adenocarcinoma, previously shown to be permissive to SARS-CoV-2 [10] was also used to validate our observations. As shown in Fig 2D, a dose-dependent decrease of viral RNA production was also observed in infected Calu-3 cells treated with clofoctol, at concentrations that did not exhibit a cytotoxic effect (Fig 2B). In this cell line, clofoctol exhibited an $IC_{50}$ of 7.9 μM.

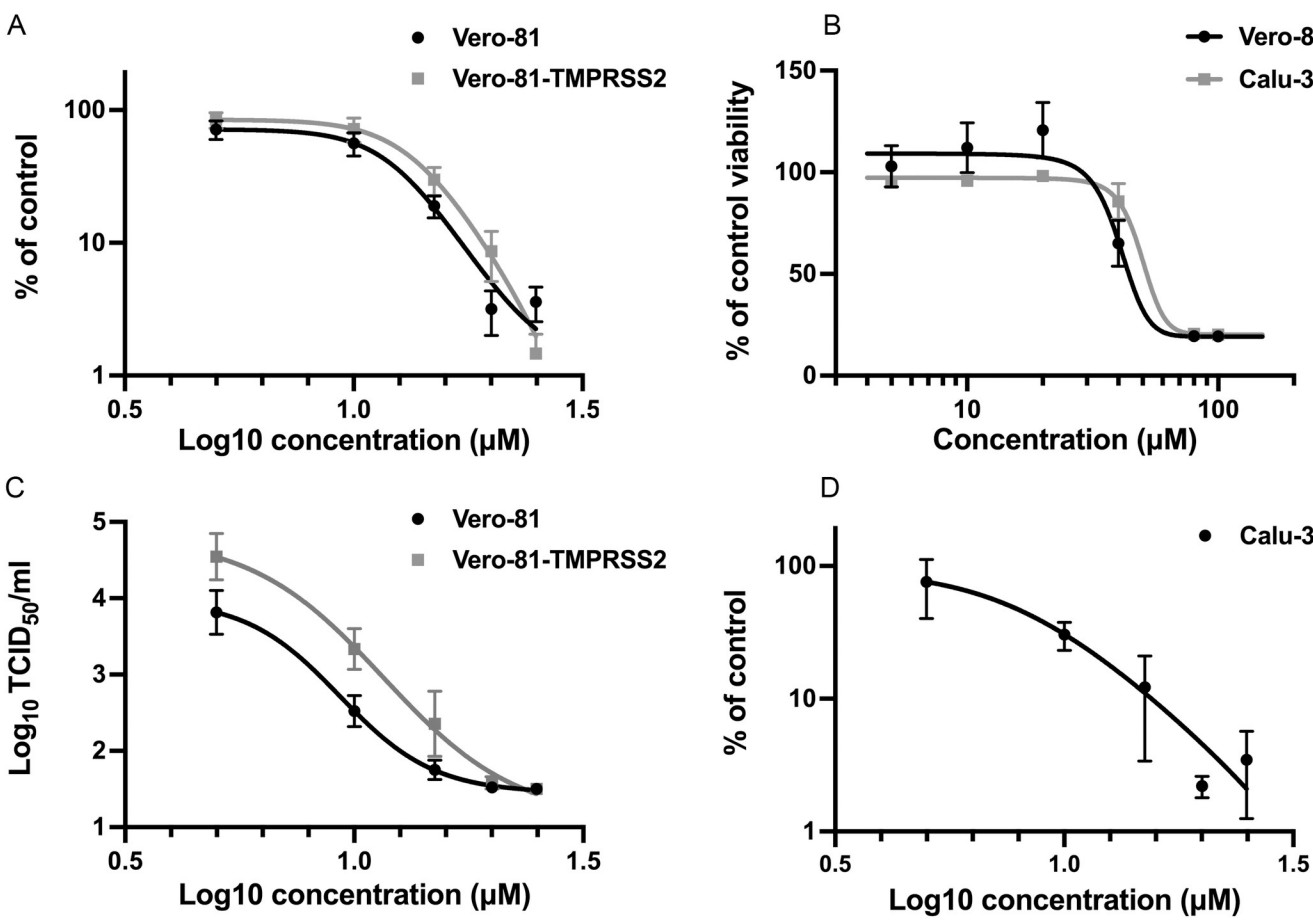

**Fig 2. In vitro validation of the antiviral activity of clofoctol. A**, Clofoctol inhibits the genomic replication of SARS-CoV-2. Vero-81 and Vero-81-TMPRSS2 cells were infected for 6h at an MOI of 0.25 in the presence of increasing concentrations of clofoctol. Then, total RNA was extracted and viral RNA was quantified by RT-qPCR and normalized by the amount of total RNA. Results are presented as the percentage of the viral load of the control and represent the average of seven independent experiments performed in duplicates. Error bars represent the standard error of the mean (SEM). **B**, Clofoctol is not cytotoxic in cell culture at concentrations below 40 μM. Vero-81 cells and Calu-3 cells were cultured in the presence of given concentrations of clofoctol. Cell viability was monitored using the MTS-based viability assay after 24 hours of incubation. **C**, Clofoctol inhibits the production of progeny virions. Vero-81 and Vero-81-TMPRSS2 cells were infected with SARS-CoV-2 at a MOI of 0.25. After 1h, the inoculum was removed and the cells were washed with PBS prior treatment with clofoctol. Cells were then further incubated for 16h. Thereafter, supernatants were collected and the amounts of secreted infectious virus were quantified. The limit of detection was 1.5TCID$_{50}$/mL. These data represent the average of three independent experiments (N = 3). Experiments were performed in duplicate for each condition. **D**, Clofoctol inhibits SARS-CoV-2 replication in Calu-3 cells. Calu-3 cells were infected at a MOI of 0.25 in the presence of increasing concentrations of clofoctol for 24h. Then, total cellular RNA was extracted and viral RNA was quantified by RT-qPCR. Results are presented as the percentage of the viral load of the control and represent the average of three independent experiments performed in duplicates. Error bars represent the standard error of the mean (SEM).

## Clofoctol inhibits the translation of SARS-CoV-2 viral RNAs

The life cycle of a virus can be divided into three major steps: (1) entry, (2) translation/replication and (3) assembly/release. To determine at which of these steps clofoctol inhibits SARS-CoV-2, the compound was added either before infection, during virus entry, post-inoculation or throughout all the steps. Remdesivir, an inhibitor of the viral polymerase [12], and CQ were used as control antivirals affecting viral replication or entry, respectively. As shown in Fig 3A, remdesivir inhibited infection only when added after the entry step, whereas CQ was only efficient when added at the entry step. Clofoctol inhibited SARS-CoV-2 mainly at the post-inoculation step, although it had also a mild effect at the entry step. These data suggest that the translation/replication step is likely the major target of clofoctol. To further characterize the

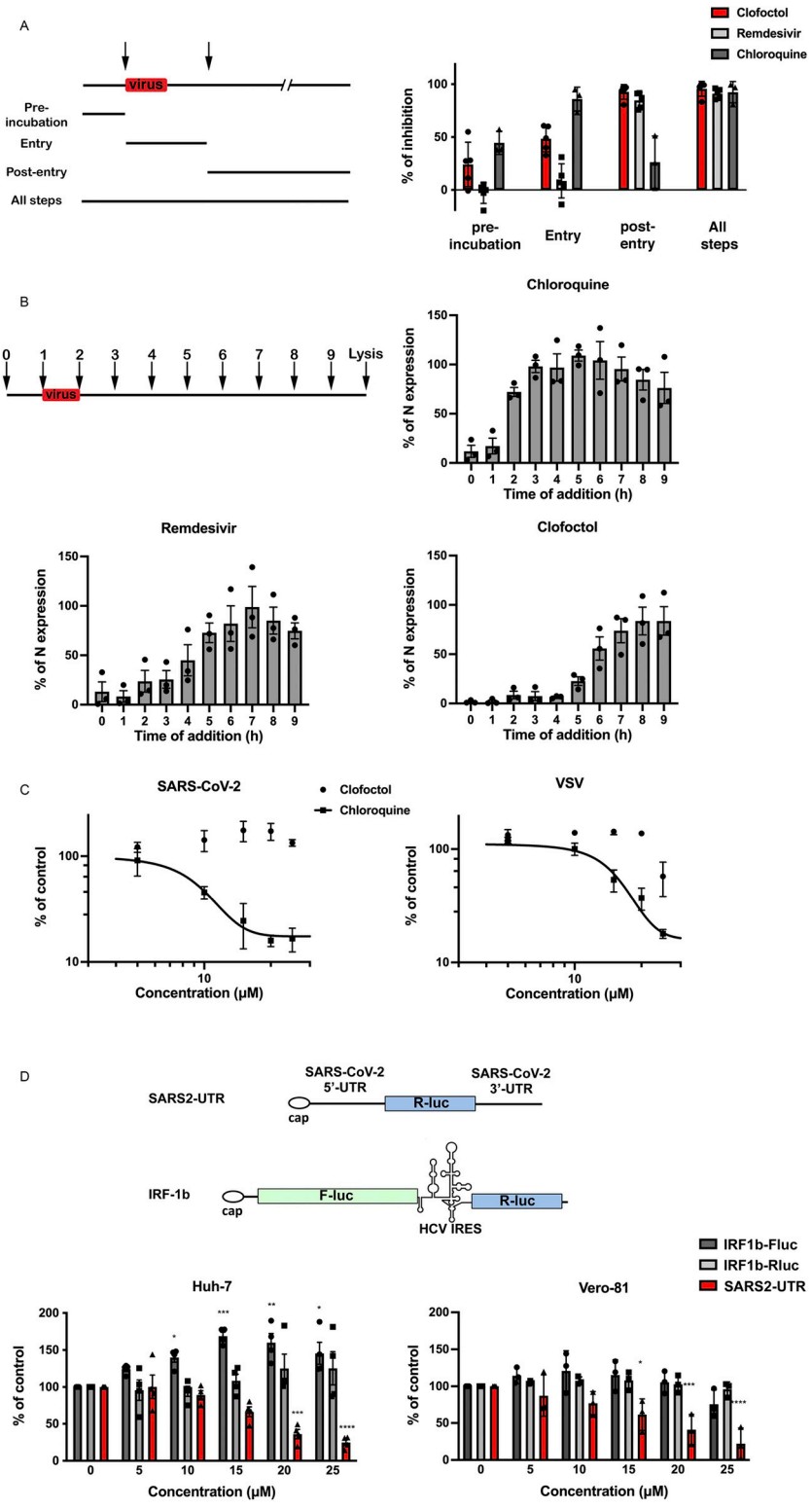

**Fig 3. Clofoctol targets the translation step of SARS-CoV-2 life cycle. A**, Clofoctol is mainly efficient at a post-entry step. Vero-81 cells were infected with SARS-CoV-2 at an MOI of 0.25. Clofoctol, remdesivir or CQ were present at a concentration of 15 μM either before infection, during virus entry, post-inoculation or throughout the steps as indicated on the schematic depiction of the experiment. Bars indicate when the drugs are present during the experiment for each condition. At 16h post-infection, cells were fixed with 4% paraformaldehyde and processed to

detect the proportion of infected cells. Therefore, they were immunostained to allow for the detection of the viral double-stranded RNA and nuclei were detected by Hoechst staining to count the total number of cells. Results are presented as the percentage of infection inhibition and represent the average of five independent experiments. **B**, Time-of-addition experiment. Vero-81 cells were infected at an MOI of 0.5 for 1h. 15 µM of clofoctol, remdesivir or CQ were added every hour starting 1h before inoculation. Cells were lysed 8h after the end of the inoculation in Laemmli loading buffer and the amount of N protein was detected in immunoblot. Results are presented as the percentage of N protein expression relative to that in non-treated cells (CTL) and represent the average of three independent experiments. Error bars represent the standard error of the mean (SEM). **C**, Clofoctol does not inhibit SARS-CoV-2 entry. Huh-7 cells expressing ACE2 receptor were infected with SARS2pp or pseudoparticles containing the envelope glycoprotein of the vesicular stomatitis virus (VSV) used as a control (VSVpp) for 3 hours in the presence of increasing concentrations of clofoctol or CQ. At 48 hours post-infection, cells were lysed to quantify luciferase activity. The results are expressed in % of the controls of three independent experiments. The experiments were performed in triplicate (n = 3) in each condition. **D**, Clofoctol inhibits viral RNA translation. Schematic representation of the reporter construct expressing the *Renilla* luciferase placed between the 5'-UTR and the 3'-UTR of the SARS-CoV-2 genomic RNA and the control bicistronic construct containing the firefly luciferase sequence under the control of a cap structure, followed by the *Renilla* luciferase under the control of hepatitis C virus (HCV) IRES. Huh-7 or Vero-81 cells were electroporated with *in vitro* transcribed RNA. Cells were lysed after 8h and luciferase activities were recorded. The results are expressed in % of the controls of three independent experiments. The experiments were performed in quadruplicate (n = 4) in each condition. Two-way ANOVA followed by the Dunnett's multiple comparisons test was performed for statistical analysis (*p < 0.05; **p < 0.01; ***p < 0.001).

post-entry inhibitory effect of clofoctol, a time-of-addition experiment was also performed in parallel with clofoctol, remdesivir and CQ. In this experiment, the compounds were added before infection or at different times post-infection. As shown in Fig 3B, CQ was only efficient when added prior to infection or during the inoculation step, whereas remdesivir and clofoctol remained effective when added up to three- to four-hours post-inoculation, respectively. To further exclude a potential effect of clofoctol on SARS-CoV-2 entry, we used retroviral particles pseudotyped with the SARS-CoV-2 Spike (S) glycoprotein (SARS2pp). These are retroviral cores carrying SARS-CoV-2 S glycoprotein in their envelope and a minigenome containing a luciferase reporter gene. In this context, only the early steps of the viral life cycle (i.e., virus interaction with receptors, uptake, and fusion) are SARS-CoV-2 specific, whereas all later steps are dependent on retroviral nucleocapsid elements. Clofoctol did not show any inhibitory effect on SARS2pp entry (Fig 3C), indicating that it does not inhibit the cellular entry of SARS-CoV-2. Of note, clofoctol was also active against another coronavirus which is mildly pathogenic, HCoV-229E, and more importantly against the D614G, B.1.1.7, B.1.351 and B1.617.2 variants of SARS-CoV-2 (Fig 4), indicating that its antiviral activity is conserved across different clades of SARS-CoV-2 and coronavirus species.

For positive-stranded RNA viruses like coronaviruses, translation of the viral genome is the step that immediately follows virus entry. To determine whether this step is affected by clofoctol treatment, we used a reporter construct expressing the *Renilla* luciferase introduced between the 5'-UTR and the 3'-UTR of the SARS-CoV-2 genomic RNA. As a control, we used a bicistronic construct containing the *Firefly* luciferase sequence under the control of a eukaryotic mRNA 5' cap structure, followed by the *Renilla* luciferase sequence under the translational control of the hepatitis C virus (HCV) IRES [13]. To avoid a potential effect of clofoctol on the transcription of the reporters from plasmid DNA, *in vitro*-transcribed capped RNAs were transfected by electroporation into Vero-81 or Huh-7 cells. After 8h of clofoctol treatment, a dose-dependent inhibition of luciferase activity was observed with the SARS-CoV-2 UTRs-based construct in Vero-81 and Huh-7 cells, but not for the control bicistronic construct (Fig 3D), indicating that clofoctol specifically inhibits the translation of an mRNA containing the UTRs of SARS-CoV-2. To determine whether global cellular protein translation is not inhibited in the presence of clofoctol, a puromycin-conjugation assay was performed to monitor de novo protein synthesis through specific C-terminal protein puromycin conjugation. As shown in S2 Fig, clofoctol did not induce an

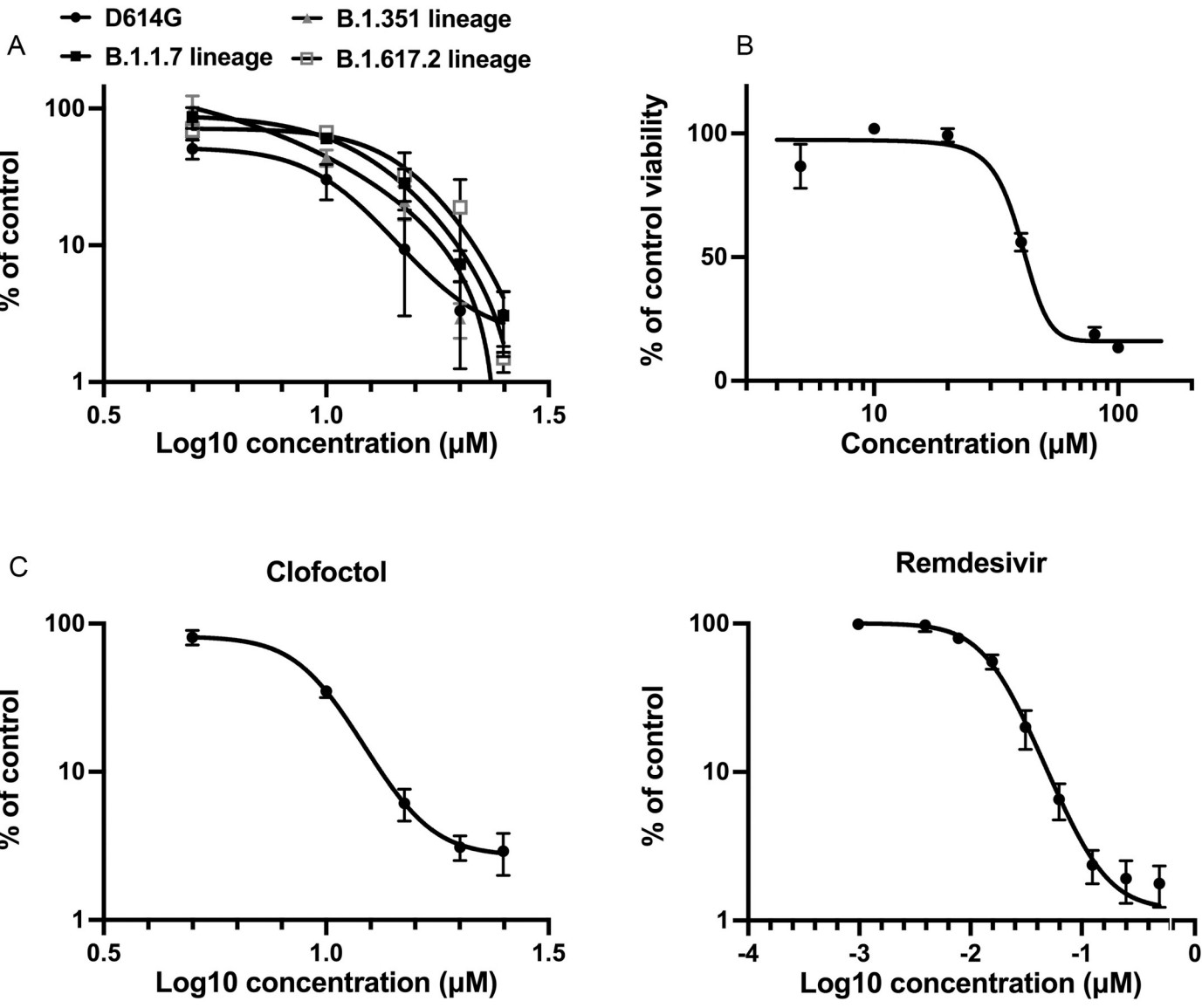

**Fig 4. Clofoctol inhibits other SARS-CoV-2 variants as well as HCoV-229E. A**, Vero-81 cells were infected either with SARS-CoV-2 of lineage B1 containing the D614G mutation (SARS-CoV-2/human/FRA/Lille_Vero-TMPRSS2/2020) or with SARS-CoV-2 of lineage B1.1.7 (GISAID accession number EPI_ISL_1653931) or lineage B.1.351 (GISAID accession number EPI_ISL_1653932) or lineage B.1.617.2 (GISAID accession number EPI_ISL_2143633). Viral genomes were quantified by RT-qPCR and normalized by the amount of total RNA. Results are presented as the percentage of the viral load of the control and represent the average of three independent experiments performed in duplicates. Error bars represent the standard error of the mean (SEM). **B**, Clofoctol is not cytotoxic in cell culture at concentrations below 40 µM. Huh-7 cells were cultured in the presence of given concentrations of clofoctol. Cell viability was monitored using the MTS-based viability assay after 24 hours of incubation. **C**, Huh-7 cells were infected with HCoV-229E-Rluc in presence of different concentrations of clofoctol or remdesivir. At 7h post-infection, cells were lysed and luciferase activities were quantified. Results are presented as the percentages of the control and represent an average of three independent experiments performed in triplicates. Errors bars represent the standard error of the mean (SEM).

inhibition of cellular protein translation. Together, these data suggests that clofoctol has the potential to inhibit the translation of genomic as well as sub-genomic SARS-CoV-2 RNAs.

## Clofoctol inhibits SARS-CoV-2 replication *in vivo* and lowers inflammation in lungs

To investigate the potential antiviral activity of clofoctol against SARS-CoV-2 *in vivo*, we took advantage of transgenic C57BL/6 mice expressing the human ACE2 receptor (K18-hACE2

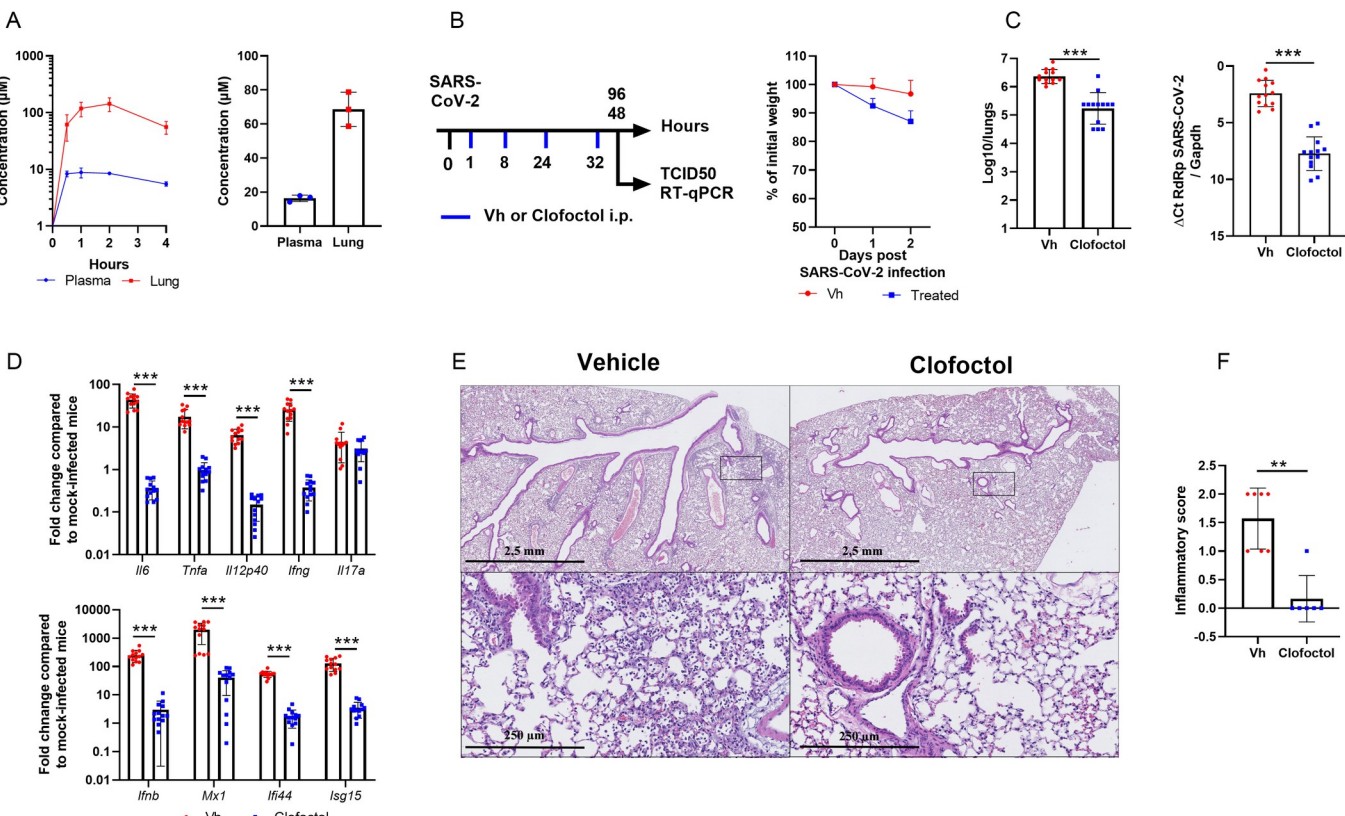

**Fig 5. Pharmacokinetics and antiviral properties of clofoctol in a mouse model of COVID-19. A**, Pharmacokinetics characterization of clofoctol in mice. *Left* panel, 8–10 week-old female C57BL/6J mice were treated i.p. with a single dose of clofoctol (62.5mg/kg) and were sacrificed at different time points thereafter. *Right* panel, Clofoctol was inoculated twice daily during two days and mice were sacrificed 1h after the last injection. Clofoctol concentrations in lungs (n = 3/time point, 3 samples/lung) and plasma (n = 3/time point, 2 technical replicates) are depicted. **B-D**, Effects of clofoctol treatment on SARS-CoV-2 infection in K18-hACE2 transgenic C57BL/6J mice. **B**, *Left panel*, Scheme of the experimental design in which the effect of clofoctol was assessed in mice. Mice were treated i.p. with clofoctol (62.5mg/kg) or vehicle 1h and 8h after i.n. inoculation of SARS-CoV-2 ($5x10^2$ TCID$_{50}$ per mouse) and treated again twice at day 1 post-infection. Animals were sacrificed at day 2 and day 4 post-infection. *Right panel*, Body weight curves are shown. **C**, The viral load was determined by titration on Vero-E6 cells (*middle panel*) and by RT-qPCR (*right panel*) (day 2 post-infection). **D**, mRNA copy numbers of genes were quantified by RT-qPCR. Data are expressed as fold change over average gene expression in mock-treated (uninfected) animals (day 2 post-infection). **E**, Lung sections were analyzed at day 4 post-infection. Shown are representative lungs (hematoxylin and eosin staining). *Lower panels*, enlarged views of the area circled in black in *upper* panels. **F**, Blinded sections were scored for levels of pathological severity. The inflammatory score is depicted. **B-C**, Results are expressed as the mean ± SD (n = 13 for panels **B** and **D** and n = 6–7 for panel **E** and **F**). Significant differences were determined using the Mann-Whitney U test (**p < 0.01; ***p < 0.001).

mice) [14]. Before testing the potential antiviral activity of clofoctol, pharmacokinetic experiments were performed in female C57BL/6 mice. To this end, clofoctol was injected intraperitoneally (i.p.) at 62.5 mg/kg to reach a lung concentration close to that achieved in humans at approved posology. Mice were sacrificed at 30 min, 1h, 2h and 4h after i.p. administration of clofoctol. As early as 30 min after injection, clofoctol reached concentrations up to 61 µM in the lungs and remained above this level for almost 4h (Fig 5A, *left panel*), whereas it remained at a concentration seven times lower in the plasma. According to its expected half-life in the lungs, clofoctol concentration was anticipated to remain above its *in vitro* measured IC$_{50}$ (IC$_{50}$ ~10µM) for more than 7 consecutive hours. It was then decided to treat the mice twice daily to maintain a lung concentration close to 60 µM. In this setting, clofoctol concentration reached 67 µM in the lungs, 1h after the fourth administration (Fig 5A, *right panel*).

Because of this favorable pharmacokinetic profile in mice, we decided to test clofoctol in K18-hACE2 transgenic mice. Female mice were inoculated intranasally (i.n.) with a lethal dose ($5x10^2$ TCID$_{50}$) of a clinical SARS-CoV-2 isolate. The animals were then injected

intraperitoneally with clofoctol (62.5 mg/kg) at 1h and 8h post-infection. This treatment was repeated the day after infection. Mice were sacrificed at day 2 or day 4 post-infection (Fig 5B, *left panel*). At the dose used, clofoctol induced body weight loss (Fig 5B, *right panel*). As compared to untreated animals, the infectious viral load detected in the lungs of clofoctol-treated mice was reduced by more than 1.1 $\log_{10}$ at day 2 post-infection (Fig 5C, *left panel*). Analysis of viral RNA yields by RT-qPCR confirmed the reduced viral load in clofoctol-treated animals (Fig 5C, *right panel*). The effect of clofoctol was also assessed in male K18-hACE2 transgenic mice, which have been described to be more susceptible than females to SARS-CoV-2 infection [14]. Clofoctol treatment similarly reduced the viral load in the lungs of male K18-hACE2 transgenic mice at a lower dose (50 mg/kg) (S3A Fig). Of note, the effect was dose-dependent (S3B Fig). We then investigated whether the decrease in viral load would have positive effects on lung inflammation. Remarkably, the expression of transcripts encoding IL-6, TNFα, IL12p40, IFNβ, IFNγ and the interferon-stimulated genes (ISG) Mx1, Ifi44 and ISG15 was markedly reduced in clofoctol-treated mice, in contrast to that of IL-17A (Fig 5D). Similar results were also observed in male K18-hACE2 transgenic mice (S3C Fig). At day 4 post-inoculation, relative to controls, female mice treated during the first 2 days with clofoctol still showed a significant decrease in viral load, albeit lower than at day 2 post-infection (S4A Fig). A decreased expression of transcripts encoding inflammatory markers was also evidenced (S4B Fig). At this time point, SARS-CoV-2 infection was associated with reduced expression of genes encoding markers of epithelial barrier function, including the tight-junction protein Zonula Occludens-1 (ZO-1) and occludin. Interestingly, the decrease in these transcript levels induced by SARS-CoV-2 infection was significantly reduced in clofoctol-treated animals (S4C Fig).

Lastly, we assessed the impact of clofoctol treatment on lung pathology at day 4 post-infection. In vehicle-treated animals, a mild multifocal broncho-interstitial pneumonia was observed (Fig 5E, *upper left panel*). Signs of moderate inflammation, with the presence of neutrophils, macrophages and a few lymphocytes, were observed within alveolar lumens, inter-alveolar septa, and perivascular spaces which was accompanied by minimal perivascular edema (Fig 5E, *lower left panel*). Slight vascular congestion and discrete intra-alveolar hemorrhages were also detected (not shown). In stark contrast, only a minimal interstitial inflammation was observed in clofoctol-treated mice (Fig 5E, *upper right panel*), with a limited presence of macrophages and lymphocytes within inter-alveolar septa and little vascular congestion (Fig 5E, *lower right panel*). Analysis of pathological scores showed statistical differences between vehicle-treated and clofoctol-treated animals (Fig 5F). We conclude that, at doses that produce lung concentrations close to those observed in human patients treated at the approved dose, clofoctol treatment in mice just after infection lowers SARS-CoV-2 replication and reduces lung pathological features associated with this viral infection.

## Discussion

In this study, we report the high-throughput screening of ~2,000 drugs, approved for human use, for their potential activity against SARS-CoV-2. Our data identify clofoctol as a promising antiviral candidate for the treatment of COVID-19 patients. This antibacterial drug was developed in the late 1970s. Its efficacy has been demonstrated for the treatment of *Streptococcus pneumoniae*—the leading cause of bacterial pneumonia worldwide—and *Staphylococcus aureus* [15,16]. The drug was marketed in France until 2005 under the trade name Octofène and is still prescribed in Italy under the trademark GramPlus. Mechanistically, clofoctol inhibits bacterial cell wall synthesis and induces membrane permeabilization [17,18]. Along with its bactericidal activity, clofoctol was recently shown to also inhibit protein translation and to

impair tumor cell growth [19,20]. As such, clofoctol could be useful to treat some cancers and possibly other diseases [21].

Among the 2,000 drugs tested, clofoctol emerged as the most promising compound to inhibit SARS-CoV-2 replication in our experimental settings. Our data show that it can contribute to inhibition of SARS-CoV-2 propagation by blocking translation of viral RNA. However, we cannot exclude other effects of clofoctol on SARS-CoV-2 replication. The inhibition of translation by clofoctol could be due to the activation of the unfolded protein response (UPR) pathways. Clofoctol has indeed been reported to induce endoplasmic reticulum (ER) stress and to activate all three UPR pathways, i.e. the inositol requiring enzyme 1 (IRE1), the double stranded RNA-activated PK-like ER kinase (PERK), and the activating transcription factor 6 (ATF6) [22]. Although UPR activation is observed during SARS-CoV-2 infection [23], chemical activation of UPR by thapsigargin has been shown to inhibit coronavirus replication, including SARS-CoV-2 [24]. Furthermore, modulating the PERK-eIF2α pathway can inhibit the replication of the transmissible gastroenteritis porcine coronavirus [25]. Similarly, triggering the UPR with 2-deoxy-D-glucose inhibits the replication of another coronavirus, the porcine epidemic diarrhea virus [22]. Whether the clofoctol-induced inhibition of SARS-CoV-2 translation is linked to UPR activation will be the focus of further investigation.

Translation of viral RNA requires interaction of viral RNA with the host cell translational machinery. Several studies have uncovered RNA binding proteins (RBPs) important for infection. It is likely that clofoctol may act on one of these RBPs to inhibit viral translation. Comparison of translational factors recruited by SARS-CoV-2 or flavirirus (ZIKV, DENV) showed different preferences for elongation initiation factors. Indeed SARS-CoV-2 prefers EIF3B, 4H, 4B, 3F, and A3, whereas flaviviruses prefer EIF3A, 4G1, 3C, and 3D [26]. More specifically, CSDE1 has been identified as a proviral factor that binds viral RNAs [27]. This is particularly interesting as clofoctol has been shown to bind CSDE1 [19]. However, whether CSDE1 binds the UTR region or another region of the viral RNA will need further investigation.

Previous pharmacokinetic studies indicate that clofoctol is well absorbed by rectal administration, and can rapidly expose lung tissues [11,28]. Of interest, as early as 90 minutes after rectal administration, the peak concentration of clofoctol that can be achieved in human lungs is more than 20 times higher than its $IC_{50}$ measured in Vero-81 cells. In our experimental conditions, clofoctol was also detected in mouse lungs at a peak concentration reaching approximately tenfold its $IC_{50}$. Notably, upon two days of treatment with doses allometrically similar to those approved for human treatment, its concentration in the lungs remained far above the $IC_{50}$ measured *in vitro*. Importantly, we demonstrate here that clofoctol treatment decreased the viral load in the lungs and drastically reduced pulmonary inflammation. These *in vivo* data, as well as the rapid onset of action expected in human pharmacokinetics, strongly support clofoctol as a therapeutic candidate for the treatment of COVID-19 patients.

In our study, K18-hACE2 transgenic C57BL/6J mice were used. Although this model is useful to evaluate the efficacy of antivirals, it has some limitations. While it is much better tolerated in humans, clofoctol induces weight loss in mice, which is attributed to a decrease in gastric emptying. Moreover, SARS-CoV-2-infected mice die from encephalitis, a disease evolution not encountered in humans. It is noteworthy that, in our experimental conditions, clofoctol failed to significantly reduce mouse mortality upon SARS-CoV-2 infection. This lack of protection against mortality is likely due to the fact that mice were only treated for 2 days to limit weight loss to a maximum of 20% (Fig 5B), whereas treatment in human patients can last for up to 10 days without specific side effects. Maintaining clofoctol treatment is likely to be required to improve efficacy in this system, as exemplified by the reduced protective effect of clofoctol on the viral load at 4 dpi (S4A Fig). Attempts are currently in progress to reduce the effect of this compound on the weight loss observed in mice.

Together with its antiviral effects, clofoctol abrogated lung inflammation. To the best of our knowledge, the anti-inflammatory effect of clofoctol has never been reported before. Together with its effect on UPR pathways, clofoctol is known to interact with different targets including (i) the Cdc7/Dbf4 protein kinase complex, which regulates the initiation of DNA replication and (ii) the upstream-of-N-Ras protein (UNR), a highly conserved RNA-binding protein known to regulate gene expression. Of interest, by binding to UNR, clofoctol activates the transcription factor Kruppel-like factor 13 (KLF13) [20], known as a tumor suppressor gene and as a regulator of T cell differentiation [29,30]. Whether the UNR/KLF13 pathway triggered by clofoctol plays a role in decreasing inflammation during SARS-CoV-2 infection deserves further investigation. Additional functional studies are urgently needed to assess the global effect of clofoctol on COVID-19 pathology.

In conclusion, the antiviral and anti-inflammatory properties of clofoctol, associated with its safety profile and unique pharmacokinetics make a strong case for proposing clofoctol as an affordable therapeutic candidate for the treatment of COVID-19 patients. Finally, the relatively low cost of this drug suggests that it is a potential clinical option for treatment of COVID-19 patients in resource poor settings.

## Methods

**Ethics statement.** All experiments involving SARS-CoV-2 were performed within the biosafety level 3 facility of the Institut Pasteur de Lille, after validation of the protocols by the local committee for the evaluation of the biological risks and complied with current national and institutional regulations and ethical guidelines (Institut Pasteur de Lille/B59-350009). The experimental protocols using animals were approved by the institutional ethical committee "Comité d'Ethique en Experimentation Animale (CEEA) 75, Nord Pas-de-Calais". The animal study was authorized by the "Education, Research and Innovation Ministry" under registration number APAFIS#25517-2020052608325772v3.

**Data reporting.** No statistical methods were used to predetermine sample size. Compounds were spotted in a randomized order on the plates during the primary screen. All the other experiments were not randomized. Investigators were blinded to allocation during the primary screen and the corresponding validation, during both assay performance and outcome assessment. For all the other assays, the investigators were not blinded.

**Cells and viruses.** Vero-81 cells (ATCC, CCL-81), Vero-E6 cells (ATCC, CRL-1586), Huh-7 cells[31] and HEK293T/17 cells (ATCC, CRL-11268) were grown at 37˚C with 5% $CO_2$ in Dulbecco's modified eagle medium (DMEM, Gibco) supplemented with 10% heat-inactivated fetal bovine serum (FBS, Eurobio). Calu-3 cells (Cliniscience, EP-CL-0054) were grown in minimum essential medium (Gibco, MEM) supplemented with glutamax (Gibco) and 10% heat-inactivated FBS.

Lentiviral vectors expressing TMPRSS2 were produced by transfection of HEK293T cells with pTRIP-TMPRSS2, phCMV-VSVG and HIV gag-pol in the presence of Turbofect (Life Technologies) according to the manufacturer's instruction. Supernatants were collected at 48h post-transfection and used to transduce Vero-E6 cells.

The BetaCoV/France/IDF0372/2020 strain of SARS-CoV-2 was supplied by the French National Reference Center for Respiratory Viruses hosted by Institut Pasteur (Paris, France). The hCoV-19_IPL_France strain of SARS-CoV-2 (NCBI MW575140) was also used for *in vivo* experiments. All SARS-CoV-2 viruses, including the variants B.1.1.7, B.1.351 and B.1.617.2 were propagated in Vero-E6 cells expressing TMPRSS2 by inoculation at MOI 0.01. Cell supernatant medium was harvested at 72h post-infection and stored frozen at −80˚C in small aliquots. All experiments were conducted in a biosafety level 3 (BSL3) laboratory.

Recombinant HCoV-229E expressing the *Renilla* luciferase was a kind gift of Dr Volker Thiel (University of Bern, Switzerland) and was propagated onto Huh-7 cells.

**Chemical libraries.** The TEELibrary was built and supplied by the APTEEUS company. It was in its version n˚4 and counted 1,942 small organic molecules approved for a use in human and selected within national and international drug repositories. It is mainly composed of active pharmaceutical ingredients (>90%) and it covers 85% of the Prestwick FDA approved collection. All molecules have been dissolved in an appropriate bio-compatible solvent (DMSO or water with adjusted pH), at a concentration compatible with the testing on living cells. The majority of them are prepared at a 10mM concentration in DMSO. CQ diphosphate was purchased from Sigma-Aldrich (Dorset, England and St. Louis, MO). CQ diphosphate was diluted to a final concentration of 10 mM in water. Clofoctol was purchased from Sigma-Aldrich (C2290) or provided directly by the manufacturer (Chiesi, Italy).

**Drug screening assay.** One day prior to infection, Vero-81 cells were seeded in black 384-well μClear plates (Greiner Bio-One), at a density of 3,000 cells per well in 30 μl DMEM, supplemented with 10% FBS and 1X Penicillin-Streptomycin solution (Gibco), using a Multi-Drop Combi Reagent dispenser (ThermoFischer Scientific). The next day, compounds from the TEELibrary were first dispensed into the 384-well plates, using an Echo 550 Liquid Handler (Labcyte). To identify the compounds of interest, they were tested at a final compound concentration that usually does not induce cytotoxicity, most of them at 15 μM. On each plate, five 3-fold serial dilutions of CQ diphosphate ranging from 0.15 μM to 15 μM were added in six replicates, as a control compound of viral inhibition (positive controls). Eleven control virus wells devoid of compound and scattered over the plate, were supplemented with 0.15% DMSO or 0.15% $H_2O$ (negative controls), respectively. Cells were infected by adding 10 μL of SARS-CoV-2 per well at a MOI of 0.01 in 10% FBS-containing medium, using a Viafill Rapid Reagent Dispenser (Integra). The plates were then incubated at 37˚ with 5% $CO_2$. At 3 days post-infection, cells were stained with 10 μg/mL Hoechst 33342 dye (Sigma-Aldrich) and 1 μg/mL PI (ThermoFischer Scientific) for 30 min at 37˚C for CPE quantification by high-content imaging.

**Dose response curves and hit validations.** The selected hits were further validated in a 6-point dose-response confirmation assay. One day prior to infection, Vero-81 cells were seeded in 384-well plates, as previously described. The next day, six 3-fold serial dilutions of compounds (0.15 to 45 μM, in duplicate) were first added to the cells. Ten μL of virus diluted in medium was then added to the wells. On each plate, twenty-six virus control wells distributed over the plates were supplemented with 0.15% DMSO and $H_2O$, respectively. CQ diphosphate was added as a control compound, at six 3-fold serial dilutions (0.15 μM to 45 μM, in duplicate). Plates were incubated for 3 days at 37˚C prior to staining and CPE quantification by high-content imaging.

**Image acquisition.** Image acquisitions were performed on a high-resolution automated confocal microscope (Opera QEHS, PerkinElmer) using a 10x air objective (NA = 0.4) for cellular infection assays. Hoechst 33342-stained nuclei were detected using the 405 nm excitation laser (Ex) with a 450/50-nm emission filter (Em). Red signals, corresponding to PI-stained nuclei from dead cells, were detected using Ex at 561 nm and Em at 600 nm. A set of 3 fields was collected from each well.

**Image-based analysis.** For total cell and dead cell detection, images from the automated confocal microscope were analyzed using multi-parameter scripts developed using Columbus image analysis software (version 2.3.1; PerkinElmer) (S1 Table). A segmentation algorithm was applied to detect nuclei labeled by Hoechst 33342 (blue) and determine total nuclei number. Briefly, a mask was first determined from input image, using the intensity threshold of Hoechst dye signal to create a region of interest corresponding to Hoechst-stained population.

The nuclei segmentation was then performed using the algorithm "Find Nuclei", as described previously [32]. Morphology properties, as area and roundness, could be used to exclude smaller objects not corresponding to nuclei. The total number of cells was quantified as Hoechst-positive nuclei. Red fluorescence signal intensities in the previous selected nuclei were quantified and used for the selection of PI positive (PI+) and negative (PI-) nuclei. Subsequently, population of dead (PI+) and viable (PI-) cells were determined. The percentage of PI + cells was calculated for each compound to select drugs having an effect on the decrease of cell death, corresponding to infection or viral replication inhibition.

**Dose-response validation in different cell lines or with different variants.** Vero-81, Vero-81-TMPRSS2 or Calu-3 cells were infected in duplicates at a MOI of 0.25 in the presence of increasing concentrations of clofoctol, ranging from 0 to 25 μM, and incubated either for 6h (Vero-81 cells) or 24h (Calu-3 cells). Then total RNA was extracted by using the Nucleospin RNA kit (Macherey Nagel) as recommended by the manufacturer. Genome quantification was performed as described [33].

**Viral secretion.** Vero-81 and Vero-81-TMPRSS2 cells were infected at a MOI of 0.25 for 1h, then the cells were rinsed 3 times with PBS and further incubated in the presence of increasing concentrations of clofoctol for 16h. Each condition was performed in duplicates. Cell supernatants were collected and viral titers were measured by the $TCID_{50}$ method.

**Pseudoparticles infection.** Retroviral Murine leukemia virus particle were pseudotyped with the SARS-CoV-2 Spike (BetaCoV/France/IDF0372/2020 strain) or the glycoprotein of the vesicular stomatitis virus (VSV-G). Briefly, HEK293T cells were co-transfected with a plasmid encoding Gag-Pol (pTG-Gag-Pol), a plasmid encoding the envelope glycoprotein and a plasmid containing a minigenome with a *Firefly* luciferase reporter gene. After 48h of incubation, cell supernatants were collected, filtered and used to transduce Huh-7 cells expressing human ACE2 in the presence of increasing concentrations of clofoctol or CQ. Transduced cells were lysed 48h later and luciferase activity was measured by using the luciferase assay system (Promega).

**Time-of-addition experiment.** Vero-81 cells were plated in 24-well plates and infected for 1h at a MOI of 0.5. Clofoctol, remdesivir or CQ were added to the cells at a concentration of 15 μM every hour starting one hour before inoculation. At 8h post-infection, the cells were lysed in non-reducing Laemmli loading buffer. Proteins were separated onto a 10% SDS-polyacrylamide gel electrophoresis and transferred on nitrocellulose membranes (Amersham). Membrane-bound N proteins were detected with a rabbit polyclonal antibody (Novus) and a horseradish peroxidase-conjugated secondary antibody (Jackson Immunoresearch). Detection was carried out by chemoluminescence (Pierce) and signals were quantified by using the gel quantification function of ImageJ. The experiment was repeated 3 times in duplicates.

**Immunofluorescence.** Vero-81 cells were plated onto glass coverslips. The day after, the cells were infected for 1h with SARS-CoV-2 at a MOI of 0.25. Clofoctol, remdesivir or CQ were added at 15 μM at different steps of the infection. The cells were either incubated 1h before inoculation (pre-incubation) or during the inoculation and for 1h after virus removal (entry step) or starting 1h after the inoculation until cell fixation (post-entry). Additional conditions with the compounds present during the whole experiment were also included as well as controls with DMSO or $H_2O$. Cells were incubated for 16h after infection and fixed with 4% paraformaldehyde. Then, cells were permeabilized for 5 min with 0.1% Triton X-100 in PBS and blocked for 30 min with 5% goat serum in PBS. Infected cells were detected by using an anti-dsRNA (J2 monoclonal antibody, Scicons) diluted in blocking buffer to detect the presence of replicating SARS-CoV-2 virus as previously determined [33]. After a 30-min incubation, cells were rinsed 3 times for 5 min in PBS and incubated for 30 min with a cyanine 3-conjugated goat anti-mouse secondary antibody (Jackson Immunoresearch) and DAPI

(4′,6-diamidino-2-phenylindole). The coverslips were rinsed with PBS 3 times for 5 min followed by a final water wash before mounting on microscope slides in Mowiol 4–88 containing medium. Images acquisitions were performed with an EVOS M5000 imaging system (Thermo Fischer Scientific) equipped with a 10X objective and light cubes for DAPI and RFP. The total number of cells was determined by counting the number of nuclei and the number of infected cells was determined by counting dsRNA-positive cells. The experiment was performed three times.

**Viability assay.** Vero cells, Huh-7 cells or Calu-3 cells were plated in 96-well plates and were then incubated the next day in 100 μl of culture medium containing increasing concentrations of clofoctol for 24h. An MTS [3-(4,5-dimethylthiazol-2-yl)-5-(3-carboxymethoxyphenyl)-2-(4-sulfophenyl)-2H-tetrazolium]-based viability assay (CellTiter 96 aqueous nonradioactive cell proliferation assay, Promega) was performed as recommended by the manufacturer. The absorbance of formazan at 490 nm is detected using an enzyme-linked immunosorbent assay (ELISA) plate reader (ELx808, BioTek Instruments, Inc.). Each measure was performed in triplicate.

**Analysis of the effect of the drug on translation.** A plasmid containing a synthetic gene encompassing the 5'-UTR (nucleotides 1–265) and the 3'UTR (nucleotides 29675–29903) of SARS-CoV-2 isolate Wuhan-Hu-1 (Genebank NC_045512.2) separated by two head-to-tail BbsI sites was produced by GeneCust. The coding sequence of *Renilla* luciferase amplified by PCR using primers containing BbsI sites was inserted between both UTRs by ligation of BbsI-restricted PCR and plasmid. In this way, the coding sequence of the luciferase was inserted between the UTRs without leaving an extra nucleotide in between. The plasmid was linearized by NsiI restriction, and the linearized DNA was then used as a template for *in vitro* transcription with the mMESSAGE mMACHINE T7 kit from Thermofischer Scientific, as recommended by the manufacturer. *In vitro*-transcribed capped RNA was delivered to Vero-81 and Huh-7 cells by electroporation. Cells were cultured for 8h in the presence of increasing concentrations of clofoctol. *Renilla* luciferase activities were measured with a *Renilla* luciferase assay from Promega. As a control, we used a bicistronic construct containing the *Firefly* luciferase sequence under the control of a cap structure, followed by the *Renilla* luciferase under the control of hepatitis C virus (HCV) IRES. *Firefly* and *Renilla* luciferase activities were measured with a dual-luciferase reporter assay system from Merck Millipore as previously reported [13].

To determine the effect of clofoctol on global protein synthesis, we used a puromycin-conjugation assay adapted from Schmidt et al.[34]. Briefly, confluent monolayers of Vero-81 or Huh-7 cells in P-24 wells were incubated with of 10 μg/ml puromycin in the presence of 0.05% DMSO, 25 μM clofoctol or 100 μM cycloheximide for 1h. Then, cells were rinsed 3 times with PBS, and lysed with 0.25 ml of SDS-PAGE loading buffer containing 2% SDS and 40 mM DTT. Lysates were incubated for 15 minutes at 70˚C and 5 μl were spotted on nitrocellulose. The blot was incubated for 1 hour in blocking solution (20 mM TrisCl pH7.4, 137 mM NaCl, 0.1% NP40, 5% non-fat dry milk). Puromycin-conjugated polypeptides were detected using anti-puromycin mAb 12D10 (purchased from Sigma-Aldrich) diluted 1:20,000 in blocking solution, followed by horseradish peroxidase-conjugated goat anti-mouse IgG secondary antibody. Chemiluminescent signals were recorded with a LAS3000 apparatus and quantified with the imageJ software using gel analysis tool.

**Pharmacokinetic study.** Clofoctol diluted in 1.75% final Kolliphor RH40 (07076, Sigma) and 1.4% final ethanol in a sodium chloride solution (0.9%) was used for intraperitoneal (i.p.) injection. The concentration of clofoctol in plasma and lungs was measured at different time points post-clofoctol injection. Plasma samples and lung tissues were collected and treated with absolute ethanol, at a ratio of 1:10 (vol/vol) and 1:50 (vol/vol), respectively. Lung tissues were homogenized with a mechanical lysis system (Tissue Lyzer II). Supernatants were

obtained by centrifugation before injection in LC-MS/MS. Samples were analysed using UPLC system Acquity I Class (Waters), combined with a triple quadrupole mass spectrometer Xevo TQD (Waters). The column, placed at 40˚C, was an Acquity BEH C8 50*2.1mm, 1.7μm column (Waters) and the following mobile phases were used: 5mM ammonium formate pH 3.75 in water, as solvent (A) and 5 mM ammonium formate pH 3.75 in acetonitrile as solvent.

**Experimental infection of K18-hACE2 transgenic mice.**   Eight week-old K18-human ACE2 expressing C57BL/6 mice (B6.Cg-Tg(K18-hACE2)2Prlmn/J) were purchased from the Jackson Laboratory. For infection, mice (both sexes) were anesthetized by i.p. injection of ketamine (100 mg/kg) and xylazine (10 mg/kg) and then intranasally infected with 50 μl of DMEM containing (or not, in a mock sample) $5x10^2$ $TCID_{50}$ of hCoV-19_IPL_France strain of SARS-CoV-2 (NCBI MW575140). Clofoctol (62.5 mg/kg in females and 10, 25, 50 and 62.5 mg/kg in males) was injected i.p. at 1h and 8h post-infection. The treatment was repeated the day after infection. Body weight was measured until day 2 post-infection. Mice were sacrificed at day 2 or day 4 post-infection.

**Determination of viral loads in the lungs of mice.**   To determine the viral loads in lungs, half of right lobes were homogenized in Lysing Matrix D tubes (MP Bio) containing 1 mL of PBS using Mixer Mill MM 400 (Retsch) (15min– 15 Hz). After centrifugation at 11,000 rpm for 5 min, the clarified supernatant was harvested for virus titration. Dilutions of the supernatant were done in DMEM with 1% penicillin/streptomycin and dilutions were transferred to Vero-E6 cells in 96-well plates for $TCID_{50}$ assay. Quantitation of viral RNA in lung tissue was performed as follows. Briefly, half of the left lobe was homogenized in 1mL of RA1 buffer from the NucleoSpin RNA kit containing 20 mM of Tris(2-carboxyethyl)phosphine). Total RNAs in the tissue homogenate were extracted with NucleoSpin RNA from Macherey Nagel. RNA was eluted with 50μL of water.

**Determination of the viral load and assessment of gene expression by RT-qPCR.**   Half of the right lobe was homogenized in 1 mL of RA1 buffer from the NucleoSpin RNA kit containing 20 mM of TCEP. Total RNAs in the tissue homogenate were extracted with NucleoSpin RNA from Macherey Nagel. RNAs were eluted with 60 μL of water.

RNA was reverse-transcribed with the High-Capacity cDNA Archive Kit (Life Technologies, USA). The resulting cDNA was amplified using SYBR Green-based real-time PCR and the QuantStudio 12K Flex Real-Time PCR Systems (Applied Biosystems, USA) following manufacturers protocol. Relative quantifications were performed using the gene coding for RNA-dependent RNA polymerase (*RdRp*) and for glyceraldehyde 3-phosphate dehydrogenase (*Gapdh*). Specific primers were designed using Primer Express software (Applied Biosystems, Villebon-sur-Yvette, France) and ordered to Eurofins Scientifics (Ebersberg, Germany). The list of primers is available in S3 Table. Relative mRNA levels ($2^{-\Delta\Delta Ct}$) were determined by comparing (a) the PCR cycle thresholds (Ct) for the gene of interest and the house keeping gene (ΔCt) and (b) ΔCt values for treated and control groups (ΔΔCt). Data were normalized against expression of the *gapdh* gene and are expressed as a fold-increase over the mean gene expression level in mock-treated mice. Viral load is expressed as viral RNA normalized to *Gapdh* expression level (ΔCt).

**Lung pathology scoring.**   Lung tissues were fixed in 4% PBS buffered formaldehyde for 7 days, rinsed in PBS, transferred in ethanol and then processed into paraffin-embedded tissues blocks. The subcontractor Sciempath Labo (Larçay, France) performed histological processing and analysis. The tissue sections in 3 μm were stained with haematoxylin and eosin (H&E) and whole mount tissues were scanned with a Nanozoomer (Hamatsu) and the morphological changes were assessed by a semi-quantitative score. For the scoring, a dual histopathology scoring system adapted from [35,36] was used to assess pulmonary changes in mice. Inflammation was scored as 0 = absent, 1 = 1–10% of lung section, 2 = 11–25% of lung section, 3 = 26–50% of lung section, and 4 = >50% of lung section affected.

**Statistical analysis.** Results are expressed as the mean ± standard deviation (SD) unless otherwise stated. All statistical analysis was performed using GraphPad Prism v6 software. A Mann-Whitney *U* test was used to compare two groups unless otherwise stated. Comparisons of more than two groups with each other were analyzed with the One-way ANOVA Kruskal-Wallis test (nonparametric), followed by the Dunn's posttest. *, $P<0.05$; **, $P<0.01$; ***, $P<0.001$.

## Supporting information

**S1 Table. Multi-parametric script used in Columbus to determine the percentage of PI + nuclei.**
(JPG)

**S2 Table. List of selected compounds.**
(XLSX)

**S3 Table. List of primers used in this study.**
(XLSX)

**S1 Fig. Representative images of the screen and SSMD values. A,** Typical images of Vero-81 cells infected with SARS-Cov-2 (top panel: MOI = 0.01) or not (lower panel: Non-infected) acquired on an OPERA QEHS High Content Screening System (PerkinElmer) and corresponding image segmentation. 1: Typical 2-color images (Blue: Hoechst label, Red: PI-label); 2 and 4: 1-color images corresponding respectively to Hoechst and PI channel images; 3: Filled green objects correspond to total segmented nuclei, 5: Filled green objects correspond to PI positive cells or dead segmented cells, 6: Circled green cells correspond to non-infected cells. **B,** SSMD values for HCS screen of the Apteeus TEELibrary for the identification of anti-SARS-CoV-2 compounds. SSMD values were calculated for all plates by comparing mean and standard deviations on both negative (Mock) and positive (Infected) controls. Dotted-line is indicative of a threshold of 3 allowing for the validation of the plates. SSMDs were calculated for both readouts.
(TIF)

**S2 Fig. Clofoctol has no effect on cellular protein translation.** Vero-81 and Huh-7 cells were incubated for 1h with 10 μg/ml puromycin in the presence of 0.05% DMSO, 25 μM clofoctol (CFT) or 100 μM cycloheximide (CHX). Then, the cells were lysed and puromycin-conjugated proteins were quantified by dot-blot with an anti-puromycin antibody. Results show the intensity of the luminescent signal in arbitrary units and represent the average of 4 independent experiments performed in triplicate. Errors bars represent the standard error of the mean (SEM).
(TIF)

**S3 Fig. Effects of clofoctol treatment in male K18-hACE2 transgenic mice. a** and **b**, Male mice were treated (50 mg/kg of clofoctol) and infected as described in Fig 4B. Mice were sacrificed at day 2 post-infection. **A,** The viral load was determined by titration on Vero-E6 cells (*left panel*) and by RT-qPCR (*right panel*). **B,** The same procedure was applied but with various doses of clofoctol (n = 7–26). **C,** mRNA copy numbers of genes were quantified by RT-qPCR (50 mg/kg of clofoctol). Data are expressed as fold change over average gene expression in mock-treated (uninfected) animals. Results are expressed as the mean ± SD (n = 5–6). Significant differences were determined using the Mann-Whitney U test (**$p < 0.01$).
(TIF)

**S4 Fig. Effects of clofoctol treatment in female K18-hACE2 transgenic mice at day 4 post-infection. A** and **B**, Female mice were treated and infected as described in Fig 4B. Mice were sacrificed at day 4 post-infection. mRNA copy numbers of genes were quantified by RT-qPCR. Panel **B** includes inflammatory genes and panel **c** includes genes involved in barrier function. Data are expressed as fold change over average gene expression in mock-treated (uninfected) animals. Results are expressed as the mean ± SD (n = 6–7). Significant differences were determined using the Mann-Whitney U test (*p < 0.05; **p < 0.01).
(TIF)

## Acknowledgments

We thank Sylvie van der Werf for sharing the SARS-CoV-2 strain BetaCoV/France/IDF0372/2020, Volker Thiel for providing HCoV-229E-RLuc and Chiesi for sharing clofoctol compound. We thank the infrastructure ChemBioFrance and the platforms ARIADNE-criblage (UMS2014-US41 PLBS) and ARIADNE-ADME to provide access to the Opera microscope and for LC-MS/MS analysis. Thanks are also due to Nathan François for technical assistance and to Imène Belhaouane, Robin Prath and Nicolas Vandenabeele for their technical help in the BSL3 facility. We are also grateful to Françoise Jacob-Dubuisson for her helpful comment on the manuscript. The immunofluorescence analyses were performed with the help of the imaging core facility of the BioImaging Center Lille Nord-de-France.

## Author Contributions

**Conceptualization:** Sandrine Belouzard, Arnaud Machelart, Valentin Sencio, Thibaut Vausselin, Priscille Brodin, Terence Beghyn, François Trottein, Benoit Deprez, Jean Dubuisson.

**Data curation:** Thibaut Vausselin, Eik Hoffmann, Alexandre Vandeputte.

**Formal analysis:** Sandrine Belouzard, Arnaud Machelart, Valentin Sencio, Thibaut Vausselin, Eik Hoffmann, Nathalie Deboosere, Yves Rouillé, Lowiese Desmarets, Karin Séron, Adeline Danneels, Loic Belloy, Camille Moreau, Alexandre Vandeputte, Julie Dumont, Florence Leroux, François Trottein, Benoit Deprez, Jean Dubuisson.

**Funding acquisition:** Benoit Deprez, Jean Dubuisson.

**Investigation:** Sandrine Belouzard, Arnaud Machelart, Valentin Sencio, Thibaut Vausselin, Eik Hoffmann, Nathalie Deboosere, Yves Rouillé, Lowiese Desmarets, Karin Séron, Adeline Danneels, Cyril Robil, Loic Belloy, Camille Moreau, Catherine Piveteau, Alexandre Biela, Séverine Heumel, Lucie Deruyter, Julie Dumont, Ilka Engelmann, Enagnon Kazali Alidjinou.

**Methodology:** Sandrine Belouzard, Thibaut Vausselin, Eik Hoffmann, Nathalie Deboosere, Yves Rouillé, Lowiese Desmarets, Karin Séron, Adeline Danneels, Loic Belloy, Camille Moreau, Catherine Piveteau, Alexandre Biela, Alexandre Vandeputte, Séverine Heumel, Lucie Deruyter, Julie Dumont, Florence Leroux, Priscille Brodin, Terence Beghyn.

**Resources:** Ilka Engelmann, Enagnon Kazali Alidjinou, Didier Hober.

**Supervision:** Florence Leroux, Didier Hober, Priscille Brodin, Terence Beghyn, François Trottein, Benoit Deprez, Jean Dubuisson.

**Validation:** Yves Rouillé, Priscille Brodin, Terence Beghyn, François Trottein, Benoit Deprez, Jean Dubuisson.

**Writing – original draft:** Sandrine Belouzard, François Trottein, Benoit Deprez, Jean Dubuisson.

**Writing – review & editing:** Jean Dubuisson.

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
