## [Decision Letter · Decision Letter 0]

16 Dec 2021

Dear Professor Dubuisson,

Thank you very much for submitting your manuscript "Clofoctol inhibits SARS-CoV-2 replication and reduces lung pathology in mice" for consideration at PLOS Pathogens. As with all papers reviewed by the journal, your manuscript was reviewed by members of the editorial board and by several independent reviewers. In light of the reviews (below this email), we would like to invite the resubmission of a significantly-revised version that takes into account the reviewers' comments.

Overall, the reviewers were positive in their examination of the manuscript. However, several points need to be addressed and added to. Specifically, reviewers 2 and 3 requested additional in vivo data, including weight loss data from the current experiments. Similarly, the addition of therapeutic treatment or a higher dose of virus would significantly strengthen the manuscript. Based on the relatively low dose, it is unclear how effective clofocotl will be in preventing severe disease following SARS-CoV-2 challenge. Similarly, therapeutic (d0 or d1) treatment would provide insight for utility in treating in humans. More exploration of the specificity to SARS-CoV-2 UTR for targeting is also encouraged. Narrowing to the 5’ or 3’UTR (or both) would strengthen the manuscript and the author should offer discussion on mechanisms of action in the discussion. While one reviewer suggested use of more current variants of interest, this request would be difficult and not necessary for this manuscript. The authors should considering moving in vitro variant data to the main text, however.

We cannot make any decision about publication until we have seen the revised manuscript and your response to the reviewers' comments. Your revised manuscript is also likely to be sent to reviewers for further evaluation.

Sincerely,

Vineet D Menachery, Ph.D.

Guest Editor

PLOS Pathogens

Michael Diamond

Section Editor

PLOS Pathogens

Kasturi Haldar

Editor-in-Chief

PLOS Pathogens

orcid.org/0000-0001-5065-158X

Michael Malim

Editor-in-Chief

PLOS Pathogens

orcid.org/0000-0002-7699-2064

Overall, the reviewers were positive in their examination of the manuscript. However, several points need to be addressed and added to. Specifically, reviewers 2 and 3 requested additional in vivo data, including weight loss data from the current experiments. Similarly, the addition of therapeutic treatment or a higher dose of virus would significantly strengthen the manuscript. Based on the relatively low dose, it is unclear how effective clofocotl will be in preventing severe disease following SARS-CoV-2 challenge. Similarly, therapeutic (d0 or d1) treatment would provide insight for utility in treating in humans. More exploration of the specificity to SARS-CoV-2 UTR for targeting is also encouraged. Narrowing to the 5’ or 3’UTR (or both) would strengthen the manuscript and the author should offer discussion on mechanisms of action in the discussion. While one reviewer suggested use of more current variants of interest, this request would be difficult and not necessary for this manuscript. The authors should considering moving in vitro variant data to the main text, however.

Reviewer's Responses to Questions

**Part I - Summary**

Reviewer #1: Effective therapeutics to counter disease caused by SARS-CoV-2 are still lacking despite the ongoing global pandemic. The authors here describe a pharmacological screen of licensed therapeutics for antiviral activity against SARS-CoV-2. From this screen they identified clofoctol, an antibacterial drug, to have high activity against SARS-CoV-2 in vivo and in vivo using a human ACE2-expressing mouse model. The design and robustness of the screening process was of high quality and included numerous biologically relevant controls to identify clofoctol as having antiviral properties regardless of SARS-CoV-2 entry route. In vitro, they demonstrate a reduction of infectious virions produced in the absence of cytotoxicity following simultaneous treatment with clofotol. Its activity again other coronaviruses and variant spike proteins from SARS-CoV-2 indicate its broader applicability as a therapeutic for coronavirus disease. Further, the authors observe a slight drop in viral load and decreased inflammatory gene expression in SARS-CoV-2-infected K18-hACE2 when treated with clofoctol. The manuscript is clearly written, the approach is appropriate, and the findings are relevant.

Reviewer #2: Belouzard et al. perform a drug screen for in their ability to inhibit SARS-CoV-2 in Vero-81 cells and obtain several hits, including the now defunct chloroquine, a parent compound of hydroxycloroquine. The investigators then focus their study on the bacterial antibiotic clofoctol. While it is unclear if these studies in mice will ultimately translate to humans and the treatments against SARS-CoV-2 remain limited and thus this study is of overall interest. Belouzard and colleagues report potentially interesting findings. However, several major points need to be appropriately addressed before this manuscript can be accepted:

Reviewer #3: Overall, this is a very nice study by Belouzard et al. with promising results for the use of clofoctol in the treatment of SARS-CoV-2. The authors identified clofoctol in a screen of 1,942 drug candidates for inhibition of CPE in Vero81 cells induced by SARS-CoV-2 infection. The IC50 of clofoctol was determined for several different cell types, and the authors show that clofoctol inhibits SARS-CoV-2 through inhibition of viral translation through a series of in vitro experiments. Importantly, the authors demonstrate antiviral efficacy of clofoctol in vivo in a K18-hACE2 mouse model of infection, showing a significant reduction in viral burden in the lungs as well as reduced induction of pro-inflammatory cytokines and inflammation in clofoctol-treated animals.

The major strengths of the study are 1) the comprehensive validation approach of the drug candidates, which include analysis of both TMPRSS2-dependent and independent entry in multiple different cell lines; 2) the identification of clofoctol as a potential therapeutic, which has not been studied to date; 3) the finding that clofoctol inhibits SARS-CoV-2 viral translation; and 4) in vivo demonstration of antiviral and anti-inflammatory impact in the K18-hACE2 mouse model. I particularly liked that the authors used live virus to perform their screen and used multiple measures of viral replication (CPE-based microscopy, RT-PCR of SARS-CoV-2 RdRp, western blot of N-levels, luciferase constructs, and TCID50), which contributes to the robustness of their findings.

**Part II – Major Issues: Key Experiments Required for Acceptance**

Reviewer #1: The following major critiques and suggestions would significantly improve the manuscript:

1. For figure 3a, when adding drugs at the various steps, it’s not clear why adding Remdsivir before virus infection has no effect. One would imagine, if it is still present at the time of infection and post-entry it would still have its effect. This is seen in figure 3b where Remdesivir is added at 0hr or 1hr. Is this lack of inhibition observed in 3a because the drug is washed off the cells during or just after infection? This is not clear from the legend and should be made explicit.

2. It seems odd that addition of clofoctol increases IRF1b Fluc activity but not the SARS-UTR luc activity? Is there a hypothesized mechanism for sequence selective translational inhibition? Later in the discussion, it is mentioned that clofoctol was shown to inhibit protein translation to impair tumor growth. It would be helpful to show that global protein translation is not inhibited in cells treated with this drug, which would otherwise indicate a cytostatic activity of this drug and imply alternative mechanisms of viral inhibition.

3. Difference in dCt RdRp over Gapdh at day 2 between clofoctol-treated vs vehicle treated mice is not maintained at day 4. This suggests that the treatment may slow viral spread early but does not completely inhibit viral spread. This observation calls into question the in vivo effectiveness of this drug to reduce the burden of infection, although the observed decrease in inflammation may reduce disease. This should be discussed by the authors.

Reviewer #2: 1. The investigators examined the antiviral activity of clofoctol in Vero-81 and in Calu-3 cells. The investigators should also examine if clofoctol can inhibit SARS-CoV-2 and specifically a relevant variant such as B.1.617.2 (delta) or B.1.1.529 (omicron) in primary human airway epithelial (HAE) cells. Currently, they investigators have only examined clofoctol against a SARS-CoV-2 isolate from June of 2020 and variants no longer relevant in Vero-81 and Vero-81 TMPRSS2 cells.

2. The investigators examine the therapeutic efficacy of clofoctol in the K18 mouse model using an early SARS-CoV-2 isolate. Notably, the effect of clofactol on SARS-CoV-2 replication is minimal. The middle panel B in Figure 4 only shows a little over 1 log in viral titer reduction in the lungs of K18 mice. The y axis should be adjusted to not overinterpret the data and should start at Log10 (1) and shown all the way to 7 – axes need to be adjusted. The investigators should also perform a clofoctol therapy mouse study using a relevant variant (either delta or omicron) in addition to the early SARS-CoV-2 isolate.

3. The K18 model is being underused and the investigators should examine if clofactol also can improve mouse survival – the investigators should perform a survival study and take the mice through day 8 or later and infect with a higher dose such as 1X10^4 or 5X10^4. The investigators are notably using a very low dose of 5X10^2 of an early SARS-CoV-2 isolate (Genbank: MW575140) and should evaluate if clofoctol can inhibit replication and mitigate disease against at least one relevant SARS-CoV-2 variant such as delta or omicron.

4. It is unclear why the authors chose 50mg/kg to treat male mice and 62.5mg/kg in females? Was a PK study performed to inform these doses? In follow up in vivo experiments, the investigators should determine if there is a dose-dependency in vivo? Can the authors pick one mouse sex and perform a low dose, a middle dose (e.g., 50 or 62.5mg/kg) and a high dose to examine if there is a dose-dependent viral inhibition by viral titer readout/immune dampening?

5. The investigators examine the viral step at which clofoctol can inhibit SARS-CoV-2 and determine a post entry step and specifically translation, by the inhibition or a reporter controlled by 5’ and 3’ UTR of SARS-CoV-2. Did the authors also sequence virus at day 4 post infection in the K18 mice? Are there, for reasons that may be unexpected/incompletely understood, selection of SARS-CoV-2 mutants arising in K18 mice treated with clofoctol compared to infected mice which are untreated with clofoctol?

Reviewer #3: Major points:

1. Can the authors include any weight loss data that was collected for the animal studies for both the infection K18-hACE2 experiments and the pharmacokinetics? This data would be helpful to further assess and support the impact of clofoctol on SARS-CoV-2 infection as well as safety and tolerability of the drug.

2. The in vivo data presented in Figure 4 demonstrates both an impact on viral replication and inflammation. While it can be difficult to unlink the impact on viral replication with induction of inflammation, the authors raise the possibility in the discussion of an anti-inflammatory role of clofoctol that may be independent from the impact on viral replication, which could be doubly useful in the treatment of severe disease when viral replication and spread may be in decline. Have the authors pushed administration of clofoctol until after peak viral load in the lungs (2-4 dpi) to determine if clofoctol may also work to dampen inflammation at these later time points?

3. The dose for the infection appears to be on the lower end of the range of infection doses given to K18-hACE2 mice in studies published by others (5 x 102 TCID50 compared to 2.5 x 103 PFU; 2.5 x 104 PFU; 103-104 TCID50). That being said, it is clear that the dose used is sufficient to induce inflammation and viral replication within the lung. Have the authors tested the efficacy of clofoctol following higher infection doses?

4. The inclusion of chloroquine and remdesivir as comparative controls that inhibit different steps in viral replication makes it very tempting to compare the efficacy of clofoctol against these two drugs. How do the IC50 of chloroquine and remdesivir in these assays compare with clofoctol?

5. The authors show that clofoctol also inhibits HCoV-229E and seems to specifically target SARS-CoV-2 transcripts but does not inhibit translation of genes following HCV IRES. Can the authors comment on if there is something specific about coronavirus replication that may confer this specificity?

**Part III – Minor Issues: Editorial and Data Presentation Modifications**

Reviewer #1: The following minor critiques would improve the manuscript:

1. For simple comparison of translation inhibition by clofoctol in the two cell lines in Figure 3d, the y-axis range of both graphs should be consistent.

2. Why are only female mice presented in figure 4? If log virus production was similar in male and female, why not pool them? Choice to separate and bury male data in the supplemental is confusing.

3. Unsure what relevance the rectal administration of clofoctol referenced in the discussion. Are the authors suggesting this would be a preferred route of administration for covid-19 patients? Have oral routes of administration been studied?

Reviewer #2: Figure 3D needs an accurate label – it should read “SARS-CoV-2-UTR” or “SARS2-UTR”. Currently, the way the label is written refers to SARS-CoV, the virus that caused the 2003 outbreak.

Reviewer #3: Minor points:

1. The authors perform a nice set of experiments to define where in the viral replication cycle that clofoctol is inhibiting SARS-CoV-2. The details regarding the first set of experiments described (Figure 3a) are missing, and specifically the time points that correspond to each step that the authors are aiming to capture.

2. How expensive is a current regimen of Octofene/GramPlus? Is it a feasible clinical option for treatment of infection in resource poor settings? I think that a brief discussion of these factors would help the reader understand the feasibility of this potential therapeutic.

PLOS authors have the option to publish the peer review history of their article (what does this mean?). If published, this will include your full peer review and any attached files.

Reviewer #1: No

Reviewer #2: No

Reviewer #3: **Yes: **Bronwyn M. Gunn
---

## [Editor Report · Decision Letter 1]

4 Apr 2022

Dear Professor Dubuisson,

We are pleased to inform you that your manuscript 'Clofoctol inhibits SARS-CoV-2 replication and reduces lung pathology in mice' has been provisionally accepted for publication in PLOS Pathogens.

Best regards,

Vineet D Menachery, Ph.D.

Guest Editor

PLOS Pathogens

Michael Diamond

Section Editor

PLOS Pathogens

Kasturi Haldar

Editor-in-Chief

PLOS Pathogens

orcid.org/0000-0001-5065-158X

Michael Malim

Editor-in-Chief

PLOS Pathogens

orcid.org/0000-0002-7699-2064

The authors have adequately addressed the major concerns of the reviewers. The added data has strengthened the overall manuscript.

One minor correction, in figure 5C the Y-axis is labeled Log10/Lung; please specify the measurement (plaque forming unit, TCID50).
---

## [Editor Report · Acceptance letter]

29 Apr 2022

Dear Professor Dubuisson,

We are delighted to inform you that your manuscript, "Clofoctol inhibits SARS-CoV-2 replication and reduces lung pathology in mice," has been formally accepted for publication in PLOS Pathogens.

Best regards,

Kasturi Haldar

Editor-in-Chief

PLOS Pathogens

orcid.org/0000-0001-5065-158X

Michael Malim

Editor-in-Chief

PLOS Pathogens

orcid.org/0000-0002-7699-2064